# Improvement of Near-Infrared Light-Emitting Diodes’ Optical Efficiency Using a Broadband Distributed Bragg Reflector with an AlAs Buffer

**DOI:** 10.3390/nano14040349

**Published:** 2024-02-12

**Authors:** Hyung-Joo Lee, Jin-Young Park, Lee-Ku Kwac, Jongsu Lee

**Affiliations:** 1CF Technical Division, AUK Corporation, Iksan 54630, Republic of Korea; lee-hyungjoo@auk.co.kr; 2Korea Photonic Technology Institute (KOPTI), Gwangju 61007, Republic of Korea; police7630@kopti.re.kr; 3Graduate School of Carbon Convergence Engineering, Jeonju University, Jeonju 55069, Republic of Korea; 4Department of Advanced Components and Materials Engineering, Sunchon National University, Suncheon 57922, Republic of Korea

**Keywords:** distributed Bragg reflector (DBR), wavelength, AlAs, buffer layer, light-emitting diode (LED)

## Abstract

This study developed an advanced 850 nm centered distributed Bragg reflector (DBR) (broadband DBR) composed of nanomaterial-based multiple structures to improve the optical efficiency of an 850 nm near-infrared light-emitting diode (NIR-LED). A combined 850 nm centered broadband DBR was fabricated by growing an 800 nm centered ten-pair DBR on a 900 nm centered ten-pair DBR (denoted as a combined DBR). The combined DBR exhibited a slightly wider peak band than conventional DBRs. Furthermore, the peak band width of the combined DBR significantly increased upon using a reflective AlAs buffer layer that reduced the overlapped reflection. The output power (20.5 mW) of NIR-LED chips using the combined DBR with an AlAs buffer layer exceeded that of a conventional 850 nm centered DBR (14.5 mW) by more than 40%. Results indicated that combining the optical conditions of wavelengths and the AlAs buffer layer effectively strengthened the broadband effect of the DBR and increased the optical efficiency of the 850 nm NIR-LED.

## 1. Introduction

In the 1900s, infrared radiation was widely used in night vision, heating, hyperspectral imaging, tracking, communications, and biological systems [1,2,3,4,5,6]. Recently, its range of applications has significantly increased following the fabrication of laser diodes (LDs) and light-emitting diodes (LEDs) for infrared wavelengths [7,8]. Near-infrared light-emitting diodes (NIR-LEDs) with a wavelength of 850 nm have become an essential light source for closed-circuit television (CCTV), remote controls, photo couplers, and infrared communications, such as infrared data associations (IrDAs) [9,10,11].

To further expand the application scope of infrared radiation, outstanding and stable optical and electronic characteristics are required to increase the output power of LEDs owing to recombination in the active region. To achieve this, nanostructure-based multiple-quantum-well (MQW) current-spreading layers and distributed Bragg reflectors (DBRs) have been developed as effective systems [12,13,14]. A DBR consists of multiple layers with alternating materials possessing distinct refractive indices. This configuration forms a structure that selectively reflects particular wavelengths of light while permitting others to pass through. DBRs can enhance the efficiency of light extraction from LEDs by providing a mirror-like surface that reflects light back into the LED material, which can improve the overall brightness and performance of LEDs. Some studies have proposed advanced DBRs using air-hybrid, metal-hybrid, and embedded nanoporous structures [15,16,17]. A DBR has been proven to significantly improve the efficiency of LEDs, and its characteristics strongly depend on interactions among its component materials. The efficiency of LEDs can be significantly improved using DBRs with higher reflectivity at a specific wavelength [18]. Although developing a DBR having a higher reflectivity at a specific wavelength is essential to LED performance improvement, only a few studies have focused on achieving a wide peak band and high reflectivity in DBRs. 

This study developed broadband DBRs composed of multiple structures of Al_x_Ga_1−x_As/Al_y_Ga_1−y_As to improve the optical efficiency of an 850 nm near-infrared LED. Twenty-pair DBRs with center wavelengths of 800, 850, 900, and 950 nm were fabricated, and two DBRs were combined by growing a ten-pair shorter-wavelength DBR directly on a ten-pair longer-wavelength DBR, followed by determining an optimal combination of DBRs (optimal combined DBRs). Finally, remarkable broadband effects were achieved by inserting an AlAs buffer layer within the optimal combined DBR (broadband DBR). The effects of the broadband DBR on the optical efficiency of the NIR-LED with a specific wavelength were experimentally verified.

## 2. Experiments

Al_x_Ga_1−x_As, GaAs, and AlAs for 850 nm NIR-LEDs with DBRs were grown on an n-type GaAs (100) substrate at a 10 ± 0.5° tilt toward [0-1-1] using a metal organic chemical vapor deposition (MOCVD) system. Trimethylgallium (TMGa) and trimethylaluminum (TMAl) were used as group III sources, arsine (AsH_3_) was used as the group V source, and disillane (Si_2_H_6_) gas and cyclo-pentadienylmagnesium (Cp_2_Mg) were used as the n- and p-doping sources, respectively. Hydrogen (H_2_) was used as the carrier gas for all sources. 

To fabricate the 850 nm NIR-LED with DBRs, n- and p-doped Al_0.2_Ga_0.8_As materials were used as the n- and p-confinement layers, respectively. The active region comprised twenty-pair MQWs, with 12 nm thick GaAs wells and 16 nm thick Al_0.15_Ga_0.85_As barriers. A 6 µm thick Al_0.3_Ga_0.7_As window layer was used to strengthen the current-spreading effect. Figure 1 shows a schematic of the 850 nm GaAs infrared LED structure used in this study. DBRs with different wavelengths were inserted between the 850 nm infrared LED and the GaAs substrate.

In this study, DBRs with different wavelengths were fabricated using an Al_1−x_Ga_x_As/Al_x_Ga_1−x_As (x = 0.9) structure. Their wavelengths were calculated as shown in Equations (1) and (2) [15]. The refractive indices of Al_0.9_Ga_0.1_As and Al_0.1_Ga_0.9_As were 3.08 and 3.57, respectively.
(1)$$h_{H}=\frac{0.25\lambda_{0}}{n_{H}cos\left(\theta_{H}\right)}$$
(2)$$h_{L}=\frac{0.25\lambda_{0}}{n_{L}cos\left(\theta_{L}\right)}$$
where *h_H_* and *h_L_* are the thicknesses of the layers, *n_H_* and *n_L_* are the refractive indices of the two materials, λ_0_ is the center wavelength determined according to the design, and *θ_H_* and *θ_L_* are the incident angles at the individual layers, respectively. In general, *θ_H_* = *θ_L_* = 0 was assumed.

Theoretically, the higher refractive index of GaAs (*n* = 3.65) and lower refractive index of AlAs (*n* = 2.99) are advantageous for the development of DBRs with high reflectivity.

However, GaAs can absorb and distort photons emitted from its active region constituting GaAs. Therefore, Al_0.9_Ga_0.1_As (low refractive index) and Al_0.1_Ga_0.9_As (high refractive index) were used for the DBR of the proposed 850 nm NIR-LEDs. In addition, DBRs with various center wavelengths were obtained by controlling the thicknesses of their layers. Reflection spectra of the developed DBRs were measured through reflection spectroscopy, with incident light wavelengths ranging from 700 to 1100 nm. Optical and electrical characteristics of the developed 850 nm GaAs infrared LEDs were analyzed using sphere-type equipment.

## 3. Results and Discussion

We prepared twenty-pair DBRs with different wavelengths (800, 850, 900, and 950 nm) to compare their reflectivities. The different center wavelengths of DBRs based on Al_1−x_Ga_x_As/Al_x_Ga_1−x_As (x = 0.9) were controlled via their thicknesses. To improve the broadband effects of DBRs, ten-pair DBRs of two different wavelengths were combined. During the combination process, a ten-pair DBR with a shorter wavelength was fabricated on a ten-pair DBR with a longer wavelength. Further, single twenty-pair DBRs were fabricated to compare their reflective and broadband properties with those of combined twenty-pair DBRs.

Figure 2 shows reflection spectra of developed DBRs at different wavelengths. As evident, these DBRs exhibited apparent reflection spectra at specific center wavelengths. A maximum reflectivity of approximately 95% was observed for all DBRs. 

DBRs at different wavelengths were experimentally combined. DBRs obtained through a combination of wavelengths constituted twenty pairs, as did DBRs with a single center wavelength. Figure 3 shows reflection spectra obtained by combining two DBRs with different wavelengths. DBRs combined with two slightly different wavelengths (800 and 850 nm, or 850 and 900 nm) exhibited relatively low reflection regions compared with those of DBRs combined with two distant wavelengths (800 and 900 nm, or 850 and 950 nm).

When two DBRs with slightly different wavelengths were combined, the center wavelength of the peak band in the combined DBR was close to that of the longer wavelength of the two-component DBR. In reflection spectra, the spectral curve of the short-component DBR appeared to be suppressed by that of the longer-component DBR. Reflection spectra in Figure 3a,b indicated similar results. However, trends observed in Figure 3c,d upon the combination of two significantly different wavelength DBRs were different from those observed in Figure 3a,b: two peak bands that were red-shifted from a short center wavelength and blue-shifted from a long center wavelength, respectively, were observed. Reflections between these two peak bands decreased owing to the overlapped reflection between the two peak bands. For the DBR with two distant wavelengths (800 nm and 900 nm) shown in Figure 3c, the peak band exhibited a slightly wider and similar reflectivity compared to that of DBRs with a single 850 nm center wavelength. However, in contrast, the two peak bands showed lower reflectivity and a steeper decrease compared to those of DBRs with a single center wavelength (shown in Figure 2d) when the combined two distant wavelengths were high (850 nm and 950 nm), as shown in Figure 3d.

Results verified that the reflectivity of combined DBRs was influenced by the distance between their wavelengths; the longer the distance between wavelengths, the wider the reflection bandwidth. However, an excessively large distance between two wavelengths that were excessively high for DBRs could result in a wider reflection loss area or a lower reflection in the two-wavelength region. These results suggested that an optimal distance existed between wavelengths and the reduction condition of optical reflection.

To further reduce the optical reflection loss between the center wavelengths in DBRs, we applied an AlAs buffer layer to the combined DBR, as shown in Figure 4. AlAs has a higher bandgap (eV = 2.16 eV) and lower refractive index (*n* = 2.99) than the Al_x_Ga_1−x_As (x = 0.1: *n =* 1.55, x = 0.9: *n* = 2.11) used to make DBRs. Therefore, photons that pass through 800 nm centered DBRs could be effectively re-reflected by the AlAs buffer and 900 and 950 nm centered DBRs for the 850 and 900 nm NIR-LEDs, respectively. We embedded a 284 nm thick AlAs buffer layer between the ten-pair low wavelength and ten-pair wavelength DBRs. 

The thickness of the AlAs buffer layer was obtained by dividing the target wavelength by its refractive index and growing it for 850 and 900 nm NIR-LEDs. Figure 4 shows the effect of the AlAs buffer layer on the reflectivity of the combined DBRs shown in Figure 3. Compared with results presented in Figure 3a, the 850 nm based peak band was blue-shifted by the additional 850 nm peak provided by the AlAs buffer layer, and the peak band was wider, as shown in Figure 4a. As shown in Figure 4b, the weak peak of the 850 nm combined DBR increased and then reached the peak of the 900 nm centered DBR in relation to results shown in Figure 3b. In particular, a distinct effect of the AlAs buffer layer was observed in the DBRs combined with two distant wavelength DBRs, as shown in Figure 4c,d. The concave region between the two peak bands was removed by the application of the AlAs buffer layer. In particular, the reflection spectrum curve presented in Figure 4c exhibited the widest and highest peak band for the 850 nm center wavelength. 

Figure 5 shows the reflection spectra curves of the single 850 nm centered DBR (twenty-pair, black squares), combined DBR with two distant wavelength DBRs (ten-pair 800 nm/ten-pair 900 nm, red circles), and broadband DBR. The broadband was derived by embedding an AlAs buffer layer into the combined DBR (green triangles).

In relation to the single 850 nm centered DBR with a peak band width of 55 nm, the combined DBR had the same reflectivity, but a slightly wider peak band (peak band width: 75 nm). With a broadband DBR, an extraordinarily wide peak band (150 nm) and high reflectivity (85–95%) were obtained. This result clearly indicated that the combination of two distant wavelength DBRs and an AlAs buffer layer can significantly improve optical reflectivity while presenting an alternative to improve the optical efficiency of 850 nm NIR-LEDs. To more effectively compare results presented in Figure 2, Figure 3, Figure 4 and Figure 5, peak band ranges (range widths) and reflectivities according to types of DBRs are summarized in Table 1. The results demonstrated that the combination of two distant wavelength DBRs with an AlAs buffer layer had the widest peak band range.

Figure 6 shows the structural schematics of the three DBRs described in Figure 5. All DBRs were located between the n-confinement (N-Al_0.2_Ga_0.8_As) and the N-GaAs (substrate). Single DBRs were grown on an N-GaAs substrate using twenty-pair AlGaAs/AlGaAs with an 850 nm center wavelength, as shown in Figure 6a. Figure 6b presents the structure of the combined DBRs with two distant wavelengths (800 and 900 nm). 

Combined DBRs were fabricated by directly growing ten-pair DBRs with a shorter wavelength onto ten-pair DBRs with longer wavelengths. Figure 6c shows the broadband DBRs structure for improving the optical efficiency of the 850 nm NIR-LED. Broadband DBRs were obtained using an AlAs buffer layer between the upper 800 nm and the lower 900 nm centered DBRs. Structural schematics of the three developed DBRs indicate that the fabrication of all DBRs was simple and could be performed in situ. 

Finally, we experimentally verified the effect of the broadband DBR on the optical efficiency of 850 nm NIR-LED chips. Figure 7 shows the light output power–current–voltage (*L–I–V*) curve of 850 nm NIR-LED chips using the three types of DBRs, as shown in Figure 6. Here, an integrating sphere (Model OPI-100 LED Electrical & Optical Test System, Withlight Company, Republic of Korea) was designed to collect scattered and emitted light from a sample in the form of a hollow sphere with a highly reflective inner surface, and was used to measure *L*–*I*–*V* characteristics of the developed LEDs. 

As shown in the *L*–*I*–*V* curve in Figure 7, the three DBRs exhibited similar voltage–current (*V–I*) characteristics. This was because they had similar thicknesses, component materials, and doping levels [19]. However, the power–current (*L*–*I*) curves of the NIR-LED chips clearly exhibited the effects of the broadband DBR. Among the 850 nm LED chips using the three types of DBRs, the highest output power was observed for the broadband DBR, followed by the combined and single 850 nm centered DBRs, respectively. In particular, the highest output power of the NIR-LED chip using the broadband DBR (20.5 mW) was 44.37% higher than that of the single 850 nm centered DBR (14.2 mW). These results confirmed that the broadband effect of DBRs can effectively improve the light output power of an NIR-LED chip. 

To obtain more detailed information, the radial thetas (half angles) of photometric values were investigated for single 850 nm centered, combined, and broadband DBRs; results are shown in Figure 8. A light distributor (Model OPI-305 Gonio-Photometer System, Withlight Company, Republic of Korea) was used to measure the intensity of the light reflected from the surface of an object at various angles and to analyze the direction and distribution characteristics of light from the light source, lighting fixture, medium, and surface. We used this equipment to measure photometric radial theta values of the developed LEDs. As evident, the radial theta of the LED chip with the single DBR (NIR-LED A) exhibited a narrower range of angles, with a high photometric value (−20°–20°) and a lower maximum photometric value (60–65.7%), than those of the other two LED chips did. Further, 60–70% of the photometric value was observed between −20°–20° of radial theta, and it steeply decreased over 30°. For the LED chip using combined DBRs (NIR-LED B), the range of the angle (−30°–25°) and the maximum photometric value (71.9–65%) were slightly increased in relation to those of NIR-LED A. 

For the radial theta of the LED chip with the broadband DBR (NIR-LED C), the maximum photometric value (72%) was higher than those of the other LED chips, and the range of its angle (−40°–40°) was considerably wider.

Compared with NIR-LED A, NIR-LED C exhibited a more than two times wider range of the angle and a 17.6% higher maximum photometric value, thereby verifying the superiority of the broadband-DBR-based NIR-LED.

## 4. Conclusions

This study developed broadband DBRs to improve the optical efficiency of an 850 nm NIR-LED. DBRs with wavelengths of 800, 850, 900, and 950 nm were fabricated, and two DBRs were mutually combined to obtain a twenty-pair DBR. The broadband effect of the developed DBRs was investigated on the basis of their reflection spectra curves. Reflection spectra curves of combined DBRs indicated a strengthened broadband effect for the combined DBR using two distant wavelength DBRs. Further, the reflection loss between the two peak bands in the combined DBR was significantly improved upon the use of the AlAs buffer layer. In terms of the reflection peak band, the broadband DBR exhibited a wider peak band (150 nm) than the single (55 nm) and combined DBRs (75 nm) did, with a high reflectivity range (85–95%). This was attributed to the reduction in the overlapped reflection loss between the two peak bands caused by the optical reflection of the AlAs buffer layer. Based on the *L*–*I*–*V* curves of the NIR-LED using single, combined, and broadband DBRs, we experimentally verified that the output power (20.5 mW) of the broadband DBR was 41.38% and 24.24% higher than those of the single DBR (14.5 mW) and combined DBR (16.5 mW), respectively. Furthermore, the NIR-LED chip with the advanced DBR exhibited a considerably wider range of angles with high photometric values (−40°–40°) and a higher maximum photometric value (72%) than those of the other DBRs.

These results confirmed that combining the optical condition of wavelengths and the AlAs buffer layer can effectively strengthen the broadband effect of DBRs and significantly increase the optical efficiency of 850 nm NIR-LEDs.

## Figures and Tables

**Figure 1 nanomaterials-14-00349-f001:**
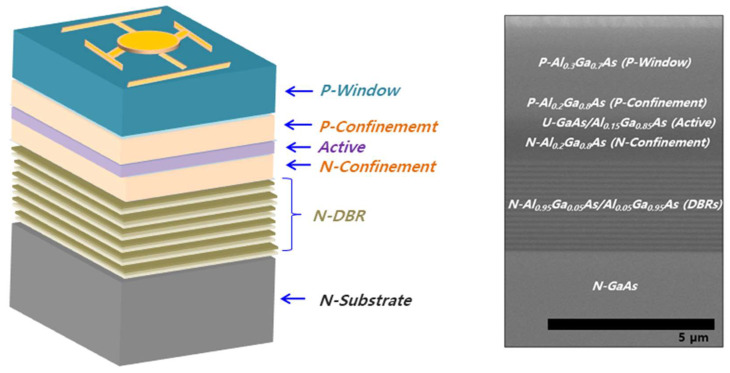
Structural schematic and scanning electron microscope image of distributed Bragg reflector (DBR) located in an 850 nm near-infrared light-emitting diode.

**Figure 2 nanomaterials-14-00349-f002:**
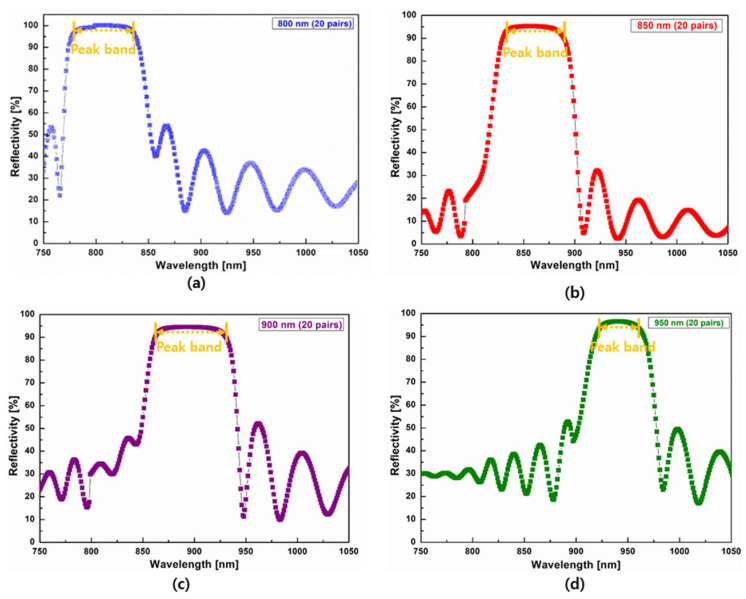
Reflection spectra of twenty-pair distributed Bragg reflectors with different center wavelengths: (**a**) 800 nm, (**b**) 850 nm, (**c**) 900 nm, and (**d**) 950 nm.

**Figure 3 nanomaterials-14-00349-f003:**
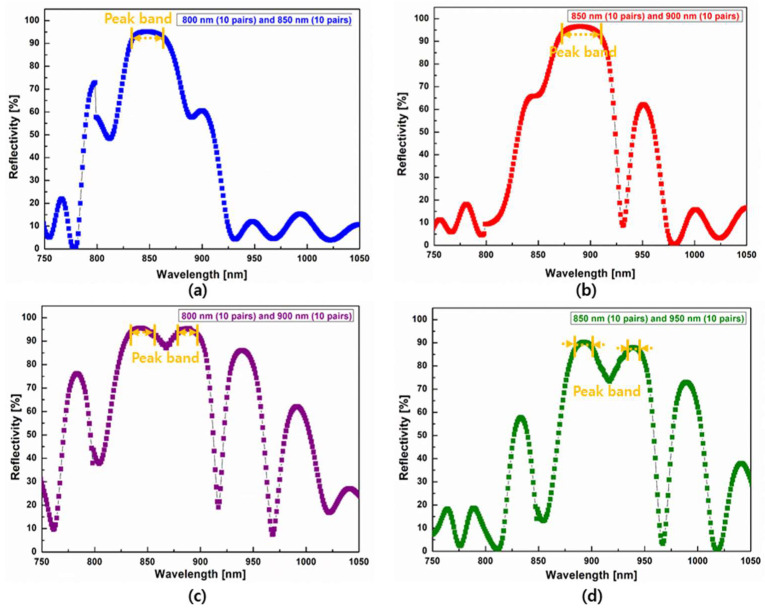
Reflection spectra of developed distributed Bragg reflectors with two close wavelengths (10 pairs + 10 pairs) and two distant wavelengths (10 pairs + 10 pairs): (**a**,**b**) two close wavelengths of 800 and 850 nm (**a**) and 850 and 900 nm (**b**); and (**c**,**d**) two distant wavelengths of 800 and 900 nm (**c**) and 850 and 950 nm (**d**).

**Figure 4 nanomaterials-14-00349-f004:**
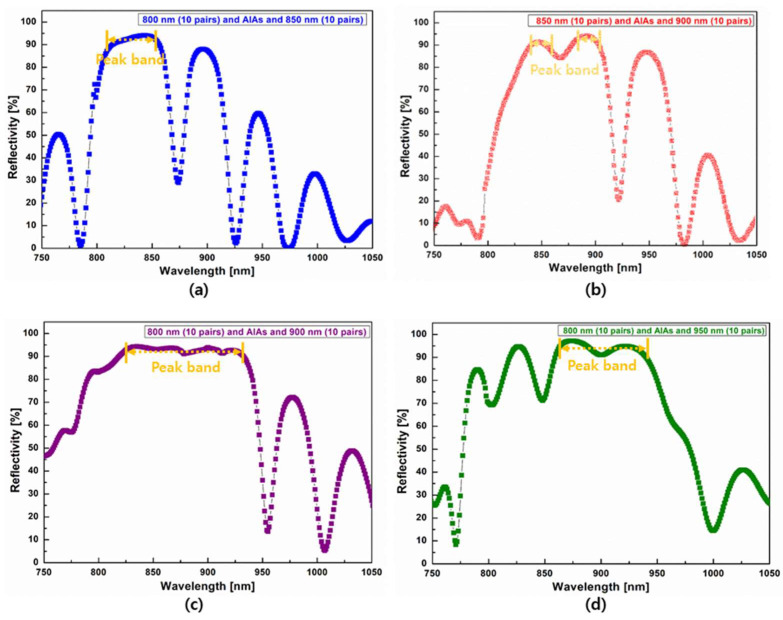
Reflection spectra of developed distributed Bragg reflector with two close wavelengths (10 pairs + 10 pairs) or two distant wavelengths (10 pairs + 10 pairs) by applying AlAs buffer layer: (**a**,**b**) two close wavelengths of 800 and 850 nm (**a**) and 850 and 900 nm (**b**); and (**c**,**d**) two distant wavelengths of 800 and 900 nm (**c**) and 850 and 950 nm (**d**).

**Figure 5 nanomaterials-14-00349-f005:**
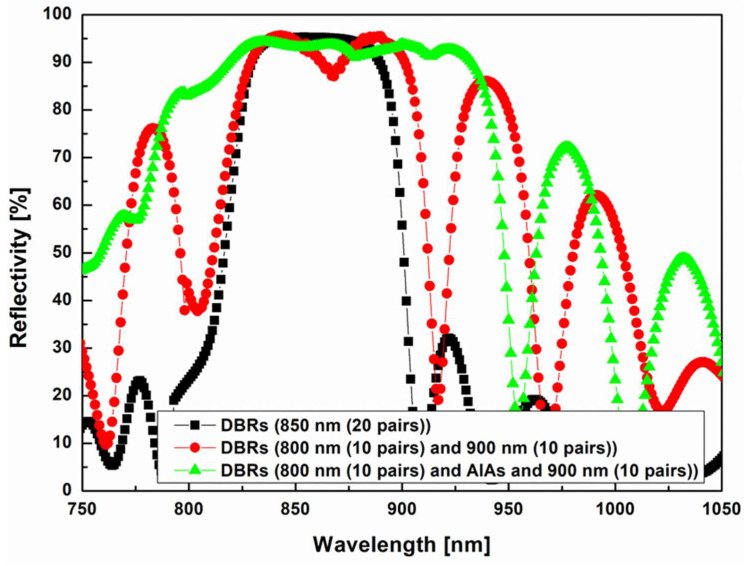
Reflection spectra of single distributed Bragg reflector (DBR), DBR with two distant wavelengths, and DBR with two distant wavelengths and AlAs buffer layer at 850 nm center wavelength.

**Figure 6 nanomaterials-14-00349-f006:**
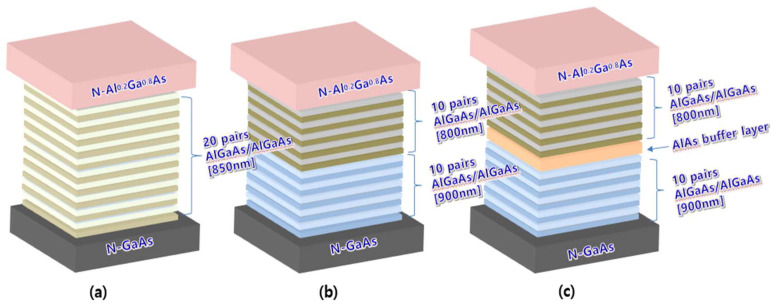
Structural schematic of (**a**) single distributed Bragg reflector (DBR), (**b**) DBR with two distant wavelengths, and (**c**) DBR with two distant wavelengths and AlAs buffer layer at 850 nm center wavelength.

**Figure 7 nanomaterials-14-00349-f007:**
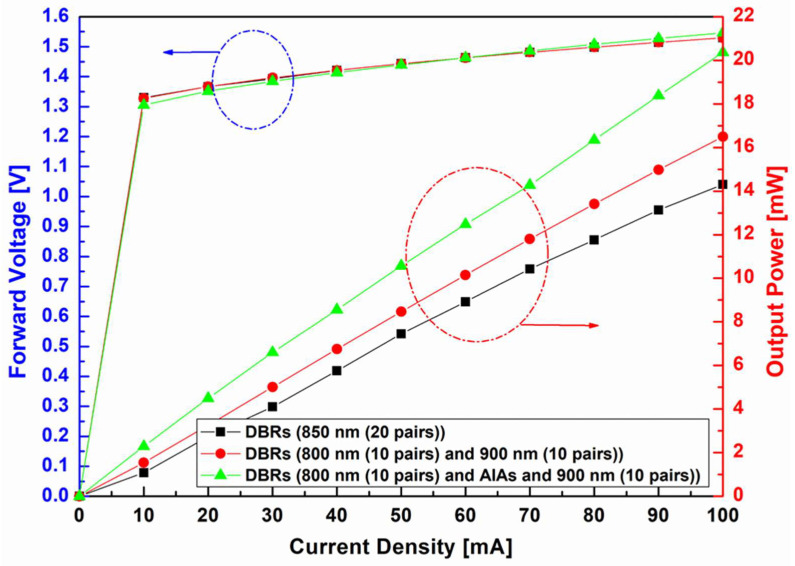
*L–I–V* characteristics curves of developed near-infrared light-emitting diode chips with single distributed Bragg reflector (DBR), DBR with two distant wavelengths, and DBR with two distant wavelengths and AlAs buffer layer. The y-axes of the graphs marked by the blue and red dotted circles represent Forward voltage [V] and Output Power [mW], respectively.

**Figure 8 nanomaterials-14-00349-f008:**
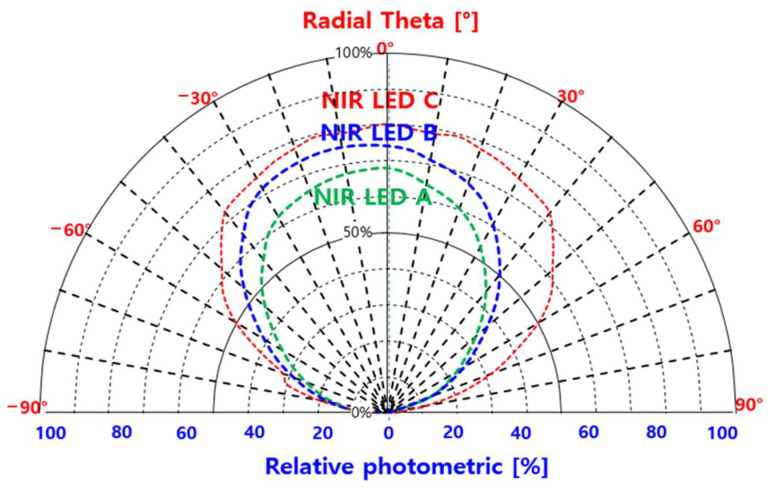
Relative photometrics and radial thetas (half angles) of near-infrared light-emitting diode (NIR-LED) A with single distributed Bragg reflector, NIR-LED B with two distant wavelengths, and NIR-LED C with two distant wavelengths and AlAs buffer layer.

**Table 1 nanomaterials-14-00349-t001:** Reflectivities and peak ranges (range widths) for a single distributed Bragg reflector (DBR), various DBRs with two distant wavelengths, and various DBRs with two distant wavelengths and an AlAs buffer layer at an 850 nm center wavelength.

DBR Types	Peak Band Range (Range Width)	Reflectivity
850 nm (20 pairs)	830–890 nm (~50 nm)	~96%
800 nm (10 pairs) + 850 nm (10 pairs)	840–860 nm (~20 nm)	~96%
850 nm (10 pairs) + 900 nm (10 pairs)	880–915 nm (~35 nm)	~96%
800 nm (10 pairs) + 900 nm (10 pairs)	840–900 nm (~60 nm)	~96%
850 nm (10 pairs) + 950 nm (10 pairs)	890–950 nm (~60 nm)	~96%
800 nm (10 pairs) + AlAs + 850 nm (10 pairs)	812–962 nm (~50 nm)	~96%
850 nm (10 pairs) + AlAs + 900 nm (10 pairs)	850–900 nm (~50 nm)	~96%
800 nm (10 pairs) + AlAs + 900 nm (10 pairs)	825–925 nm (~100 nm)	~96%
850 nm (10 pairs) + AlAs + 950 nm (10 pairs)	870–925 nm (~55 nm)	~96%

## Data Availability

Data are contained within the article.

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
