# Peer review of "Improvement of Near-Infrared Light-Emitting Diodes’ Optical Efficiency Using a Broadband Distributed Bragg Reflector with an AlAs Buffer"

_nanomaterials, 2024, doi:10.3390/nano14040349_

Round 1

Reviewer 1 Report

Comments and Suggestions for Authors

I agree with the authors that the reflection spectrum curves of the combined DBRs indicated an improved broadband effect for the combined DBR using two DBRs for far wavelengths and that the reflection loss between the two peak bands in the combined DBRs was significantly improved by the AlAs buffer layer. However, the efficiency of the developed LEDs is significantly lower than that of commercial AlGaAs LEDs emitting at the same wavelength. I found that the output power of the commercial LED is 26 mW at 50 mA, while the LED prepared by the authors is only 11 mW at the same current. The only conclusion from this is that the LEDs prepared by the authors are significantly worse than commercial ones.

Author Response

Comments and Suggestions for Authors

I agree with the authors that the reflection spectrum curves of the combined DBRs indicated an improved broadband effect for the combined DBR using two DBRs for far wavelengths and that the reflection loss between the two peak bands in the combined DBRs was significantly improved by the AlAs buffer layer.

[Comment #1]

However, the efficiency of the developed LEDs is significantly lower than that of commercial AlGaAs LEDs emitting at the same wavelength. I found that the output power of the commercial LED is 26 mW at 50 mA, while the LED prepared by the authors is only 11 mW at the same current. The only conclusion from this is that the LEDs prepared by the authors are significantly worse than commercial ones.

[Response]

We appreciate the reviewer for this reasonable comment. Advanced techniques such as metal reflector and curved surface texture are used in commercial LEDs with high power. However, the techniques have known or unknown side effects [1]. The main contribution of our study is to develop a broadband DBR by combining two distant wavelength DBRs and by using AlAs buffer layer and to verify its performance. Therefore, we have used simple AlGaAs LED in this study in spite of its low output power.

[1] Wang, T.; Zhang, X.; Liu, Y.; Chong, W.; Huang, Z.; Lu, Z.; Zhang, X.; Shi, W.; Wang, Q.; Zeng, Z.; Zhang, B.; GaN on Si micro resonant-cavity light-emitting diodes with dielectric and metal mirrors, Optic. Mater. 2023, 143, 114096.

Reviewer 2 Report

Comments and Suggestions for Authors

The manuscript titled "Improvement of optical efficiency of near-infrared light emitting diodes using broadband distributed Bragg reflector with AlAs buffer," focusing on the innovation of new device structure for high performance of NIR LED. By using a well design structure of distributed Bragg reflectors (DBRs) with AlAs, the optical efficiency of NIR-LED has been dramatically increased. The authors have provided a clear explanation of the experimental process and corresponding data. The methodology appears be to sound and the result indicate a promising outcome for the field of nanotechnology and photonics device.

In conclusion, I would like to suggest to accpet this manuscript for publication. However, I would like to give some suggestions to make this paper more readable and straight forward.

1.      The introduction could benefit from a more detailed explanation of Distributed Bragg Reflectors (DBRs) and a discussion of recent advancements in the field. This would provide readers with a stronger foundation for understanding the significance of the study.

2.      The presentation of the data in Figures 2 through 5 could be streamlined for better clarity. Currently, Figure 5 redundantly overlays data from Figures 2, 3, and 4, which themselves display very similar information. A more concise presentation, such as combining these figures into one and summarizing the detailed data in a table or histogram, would enhance the readability and eliminate redundancy.

Comments on the Quality of English Language

In the manuscript, I've noticed an inconsistency in tense usage during the description of data, which can be confusing to readers. I also noticed that certain abbreviations which typically require italicization for clarity and to adhere to scientific writing standards are not formatted as such.

Please check those issues.

Author Response

Comments and Suggestions for Authors

The manuscript titled "Improvement of optical efficiency of near-infrared light emitting diodes using broadband distributed Bragg reflector with AlAs buffer," focusing on the innovation of new device structure for high performance of NIR LED. By using a well design structure of distributed Bragg reflectors (DBRs) with AlAs, the optical efficiency of NIR-LED has been dramatically increased. The authors have provided a clear explanation of the experimental process and corresponding data. The methodology appears be to sound and the result indicate a promising outcome for the field of nanotechnology and photonics device.

In conclusion, I would like to suggest to accpet this manuscript for publication. However, I would like to give some suggestions to make this paper more readable and straight forward.

[Comment #1]

The introduction could benefit from a more detailed explanation of Distributed Bragg Reflectors (DBRs) and a discussion of recent advancements in the field. This would provide readers with a stronger foundation for understanding the significance of the study.

[Response]

We appreciate the reviewer for this pertinent comment. We have added more detailed explanation of DBRs and the discussion of their recent advancement with the related references ([16] – [18]) to the introduction section as follows:

A DBR consists of multiple layers with alternating materials possessing distinct refractive indices. This configuration forms a structure that selectively reflects particular wavelengths of light, while permitting others to pass through. DBRs can enhance the efficiency of light extraction from an LED by providing a mirror-like surface that reflects light back into the LED material, which can improve the overall brightness and performance of LEDs. Some studies have proposed advanced DBRs by using air-hybrid, metal-hybrid, and embedded nano-porous structures [16–18]. A DBR has been proven to significantly improve the efficiency of LEDs, and its characteristics strongly depend on the interactions among its component materials. The efficiency of LEDs can be significantly improved using DBRs with higher reflectivity at a specific wavelength [15]. Although developing a DBR having a higher reflectivity at a specific wavelength is essential to LED performance improvement, only a few studies have focused on achieving a wide peak band and high reflectivity in DBRs.

[16] Oh, H.S.; Ryu, S.H.; Park, S.H.; Jeong, T.; Kim, Y.J.; Lee, H.J.; Cho, Y.D.; Kwak, J.S.; Beak, J.H. Air-hybrid distributed Bragg reflector structure for improving the light output power in AlGaInP based LEDs. J.Nano. Nanotech. 2015, 15, 5048–5051.

[17] Lee, H.J.; Kwac, L.K. Study on combined reflectors for improving efficiency of high power near-infrared emitters. J. Lumin. 2022, 250, 119086–119090.

[18] Shiu, G.Y.; Chen, K.T,; Fan, F.H.; Huang, K.P.; Hsu, J.J.; Lai, C.F.; Lin,C.F.; InGaN light-emitting diodes with an embedded nanoporous GaN distributed Bragg reflectors. Sci. Rep. 2016, 6, 1–8.

[Comment #2]

The presentation of the data in Figures 2 through 5 could be streamlined for better clarity. Currently, Figure 5 redundantly overlays data from Figures 2, 3, and 4, which themselves display very similar information. A more concise presentation, such as combining these figures into one and summarizing the detailed data in a table or histogram, would enhance the readability and eliminate redundancy.

[Response]

We appreciate this co mment. We have added Table 1 to the revised manuscript.

Table 1. Reflectivity and peak range (range width) for a single distributed Bragg reflector (DBR), various DBRs with two distant wavelengths, and various DBRs with both two distant wavelengths and AlAs buffer layer at 850 nm center wavelength.

DBR types

Peakband range

(Range width)

Reflectivity

850 nm (20 pairs)

830 – 890 nm (~50 nm)

~96%

800 nm (10 pairs) + 850 nm (10 pairs)

840 – 860 nm (~20 nm)

~96%

850 nm (10 pairs) + 900 nm (10 pairs)

880 – 915 nm (~35 nm)

~96%

800 nm (10 pairs) + 900 nm (10 pairs)

840 – 900 nm (~60 nm)

~96%

850 nm (10 pairs) + 950 nm (10 pairs)

890 – 950 nm (~60 nm)

~96%

800 nm (10 pairs) + AlAs + 850 nm (10 pairs)

812 – 962 nm (~50 nm)

~96%

850 nm (10 pairs) + AlAs + 900 nm (10 pairs)

850 – 900 nm (~50 nm)

~96%

800 nm (10 pairs) + AlAs + 900 nm (10 pairs)

825 – 925 nm (~100 nm)

~96%

850 nm (10 pairs) + AlAs + 950 nm (10 pairs)

870 – 925 nm (~55 nm)

~96%

[Comment #3] 

Comments on the Quality of English Language :

In the manuscript, I've noticed an inconsistency in tense usage during the description of data, which can be confusing to readers. I also noticed that certain abbreviations which typically require italicization for clarity and to adhere to scientific writing standards are not formatted as such.

[Response]

We appreciate the reviewer’s comment. We have modified the manuscript according to the reviewer’s comment.

Reviewer 3 Report

Comments and Suggestions for Authors

The authors made a plain description of their results on using DBRs to improve the amount of ligth extracted by IR LEDs. The analysis is simple, but correct.

Some suggestions to eventually improve the presentation:

- Figure 6 and its description could be more effectively anticipated in order to make immediate and easier to the reader the sample configuration.

- The desciption and comment of Figures 3 and 4 is somehow difficult to follow; it could be made more clear by simplifyig it and concentrate the description on the maxima.

- I am wondering if the content of Figure 8 can e more effectively shown in terms of a single intensity vs angle plot of the three cases.

Comments on the Quality of English Language

A few very long sentences could be roken to simplify the readig.

Author Response

Comments and Suggestions for Authors

The authors made a plain description of their results on using DBRs to improve the amount of light extracted by IR LEDs. The analysis is simple, but correct.

Some suggestions to eventually improve the presentation:

[Comment #1]

Figure 6 and its description could be more effectively anticipated in order to make immediate and easier to the reader the sample configuration. The desciption and comment of Figures 3 and 4 is somehow difficult to follow; it could be made more clear by simplifyig it and concentrate the description on the maxima.

[Response]

We appreciate the reviewer’s comment. To solve this problem, we have added Table 1 to the revised manuscript.

Table 1. Reflectivity and peak range (range width) for a single distributed Bragg reflector (DBR), various DBRs with two distant wavelengths, and various DBRs with both two distant wavelengths and AlAs buffer layer at 850 nm center wavelength.

DBR types

Peakband range

(Range width)

Reflectivity

850 nm (20 pairs)

830 – 890 nm (~50 nm)

~96%

800 nm (10 pairs) + 850 nm (10 pairs)

840 – 860 nm (~20 nm)

~96%

850 nm (10 pairs) + 900 nm (10 pairs)

880 – 915 nm (~35 nm)

~96%

800 nm (10 pairs) + 900 nm (10 pairs)

840 – 900 nm (~60 nm)

~96%

850 nm (10 pairs) + 950 nm (10 pairs)

890 – 950 nm (~60 nm)

~96%

800 nm (10 pairs) + AlAs + 850 nm (10 pairs)

812 – 962 nm (~50 nm)

~96%

850 nm (10 pairs) + AlAs + 900 nm (10 pairs)

850 – 900 nm (~50 nm)

~96%

800 nm (10 pairs) + AlAs + 900 nm (10 pairs)

825 – 925 nm (~100 nm)

~96%

850 nm (10 pairs) + AlAs + 950 nm (10 pairs)

870 – 925 nm (~55 nm)

~96%

[Comment #2] 

I am wondering if the content of Figure 8 can be more effectively shown in terms of a single intensity vs angle plot of the three cases.

[Response]

We appreciate the reviewer for this pertinent comment. Figure 8 has been arranged in terms of a single type for the readers.

[Comment #3] 

Comments on the quality of English Language:

A few very long sentences could be broken to simplify the reading.

[Response]

Thank you for the feedback. We have modified the manuscript according to the reviewer’s comment to enhance the reader's understanding.

Round 2

Reviewer 1 Report

Comments and Suggestions for Authors

I agree that the main contribution of your study is to develop a broadband DBR by combining two distant wavelength DBRs and by using AlAs buffer layer and to verify its performance. You should have emphasized that you have used simple AlGaAs LED. It will be interesting what efficiency you achieve with a normal LED structure.